# Sex and neo-sex chromosome evolution in beetles

Ryan Bracewell[1,2], Anita Tran[1], Kamalakar Chatla[1], Doris Bachtrog[1]*

1 Department of Integrative Biology, University of California, Berkeley, California, United States of America,
2 Department of Biology, Indiana University, Bloomington, Indiana, United States of America

* dbachtrog@berkeley.edu

## Abstract

Beetles are the most species-rich group of animals and harbor diverse karyotypes. Most species have XY sex chromosomes, but X0 sex determination mechanisms are also common in some groups. We generated a whole-chromosome assembly of *Tribolium confusum*, which has a neo-sex chromosome, and utilize eleven additional beetle genomes to reconstruct karyotype evolution across Coleoptera. We identify ancestral linkage groups, termed Stevens elements, that share a conserved set of genes across beetles. While the ancestral X chromosome is maintained across beetles, we find independent additions of autosomes to the ancestral sex chromosomes. These neo-sex chromosomes evolve the stereotypical properties of sex chromosomes, including the evolution of dosage compensation and a non-random distribution of genes with sex-biased expression. Beetles thus provide a novel model to gain a better understanding of the diverse forces driving sex chromosome evolution.

**Data Availability Statement:** All the raw sequencing reads, draft genome assembly, and annotation for Tribolium confusum are under BioProject PRJNA715483. The genome assembly is also available at NCBI through accession JAGFVK000000000. Associated assemblies and

## Author summary

The evolution of differentiated sex chromosomes (e.g., X and Y chromosomes in male heterogametic species) from ordinary autosomes has occurred many times in animals and plants. Investigation of sex chromosomes in key model species (such as mammals, birds, flies and worms) has revealed that sex chromosome evolution is driven by both general and lineage-specific forces, but a detailed investigation and understanding of sex chromosomes from many clades is lacking. Importantly, cytogenetic studies in beetles–the most species-rich group of animals—have shown that chromosomal sex determination is conserved across Coleoptera. However, the identity and conservation of sex chromosomes in beetles at the molecular level and the evolutionary processes shaping them have not been studied in a systematic way. We use genomic approaches to understand karyotype and sex chromosome evolution across Coleoptera to establish general principles underlying the evolution of sex chromosomes in the most speciose group of animals on earth.

data to produce all plots are available at https://figshare.com/projects/Sex_chromosome_evolution_in_beetles/157650.

**Funding:** This research has been funded by grant 5R01GM101255 of the National Institutes of Health (https://www.nih.gov/) to DB. The funders had no role in study design, data collection and analysis, decision to publish, or preparation of the manuscript.

**Competing interests:** The authors have declared that no competing interests exist.

## Introduction

The order Coleoptera (beetles) is the most species-rich group of animals, with well over 400,000 described taxa. Cytogenetic investigations across thousands of beetle species have revealed a rich diversity of karyotypes. Chromosome numbers vary more than in many other insect groups, suggesting high rates of chromosomal fusions and fissions [1,2]. Cytogenetic analysis has shown that most species harbor X and Y chromosomes, but X0 sex determination is also common among some groups [3]. In particular, information on sex determination is available for over 4500 beetle species, and 93% of species reported show genetic sex determination (17% are X0 and 75% have XY sex chromosomes). Haplodiploidy and paternal genome elimination is found in only 10 species (0.2%), and parthenogenesis has been found for 7% of the species investigated [2]. In several clades, fusions between sex chromosomes and autosomes have been reported, creating so called neo-sex chromosomes [4–6]. Beetle sex chromosome karyotypes are typically inferred from testis squashes, and often reported as XY, X0, and Xy+ [2,7]. XY sex chromosomes typically undergo synapsis during meiosis and can recombine in their pseudoautosomal region in males (but note that achiasmatic male meiosis has evolved in some groups [2,7]). Xy+ sex chromosomes, on the other hand, pair at a distance with no synapsis and no opportunity for recombination during male meiosis [2,7].

Despite a rich cytogenetic literature [4], surprisingly little is known about chromosome synteny and karyotype evolution at the molecular level. In particular, it is unclear how well chromosome synteny is conserved in Coleoptera. In Drosophila, and across Diptera, ancestral linkage groups are maintained, and termed 'Muller elements' [8]. Gene content of Muller elements is highly conserved and chromosome numbers are relatively stable across flies. Whether similar chromosomal elements exist in beetles, and how broadly they are conserved across the beetle phylogeny, is unclear.

Additionally, while chromosomal sex determination is generally conserved across beetles, it is unclear whether the identity of sex chromosomes is conserved. For example, Diptera generally have XY sex chromosomes, but genomic analysis revealed tremendous hidden diversity in sex chromosome karyotypes among flies, with numerous transitions of sex chromosomes, or incorporations of autosomes into the ancestral sex chromosome [9]. The X chromosome appears to be conserved in some beetles [10–12] but this has not been explored rigorously using broad phylogenetic sampling.

Sex chromosomes follow unique evolutionary trajectories, with Y chromosomes losing most of their ancestral genes, and X chromosomes, in some taxa, evolving dosage compensation. Sex chromosomes originated from initially homologous chromosomes, and Y chromosomes degenerate due to their lack of recombination which renders natural selection less efficient [13,14]. In response to Y degeneration, mechanisms may evolve to balance reduced transcript levels of hemizygous X-linked genes in males and equalize transcript levels in males and females from the X and the autosomes (i.e. dosage compensation) [13,14]. In addition, sex chromosomes sometimes show an excess/deficiency of genes with sex-biased expression, with X chromosomes becoming enriched for female-biased genes and depleted for male-biased genes [15]. Whether dosage compensation exists broadly in beetles is unclear and has only been explored in *Tribolium castaneum* [10,16,17]. Comparative analysis is needed to address the prevalence of these patterns and the evolutionary forces driving them across species.

Here, we use genomic approaches to reconstruct karyotype evolution across Coleoptera. In particular, we generate a new chromosome-level assembly for *Tribolium confusum* and use published assemblies from eleven additional species spanning across the phylogeny, to broadly reconstruct sex chromosome and karyotype evolution in beetles (Fig 1). In addition, we collect RNA-seq data from different tissues to explore functional changes on the more recently

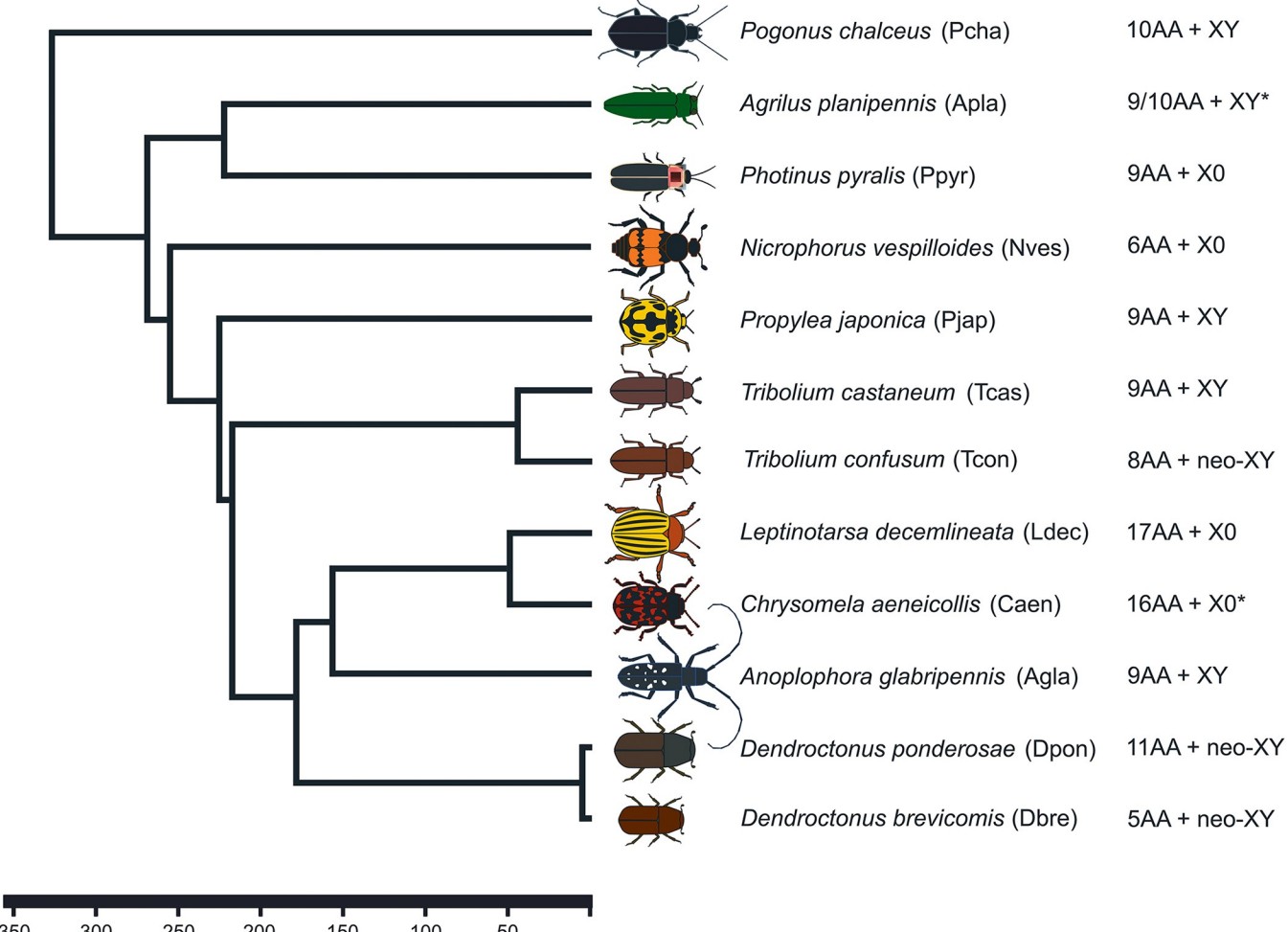

**Fig 1. Chromosome evolution in beetles.** Phylogenetic relationships of twelve beetle species used in the analysis and the evolutionary timescale and splitting events (Adapted from [62,63]). Karyotypes shown to the right. Species with an unknown karyotype are noted (*) and are given a karyotypic formula of a close relative.

formed neo-sex chromosomes in *T. confusum* and *Dendroctonus ponderosae*, allowing us to explore the evolutionary trajectory of young sex chromosomes in beetles.

## Results & Discussion

### Chromosome-level assembly of *Tribolium confusum*

We used Nanopore and Illumina sequencing combined with Hi-C scaffolding to generate a high-quality, chromosome-level assembly of the confused flour beetle, *T. confusum* (Fig 2A). The total length of the assembly is 305,133,470 bp and we were able to annotate a total of 15,231 protein coding genes (Tables 1 and S1). We found the average GC content for the genome is 30.72%. Repeat masking using our custom repeat library amounted to 147,396,050 bp (48.31%) of the draft genome being masked. Interestingly, we find the *T. confusum* genome assembly is ~140 Mb larger than the assembly of its relative *T. castaneum*, and these differences appear to be the result of an increase in repetitive elements that are broadly distributed across the genome of *T. confusum* (Fig 2B). Many repetitive elements are currently unclassified, but we find that DNA transposons make up over 10% of the genome assembly (Table 2). Our

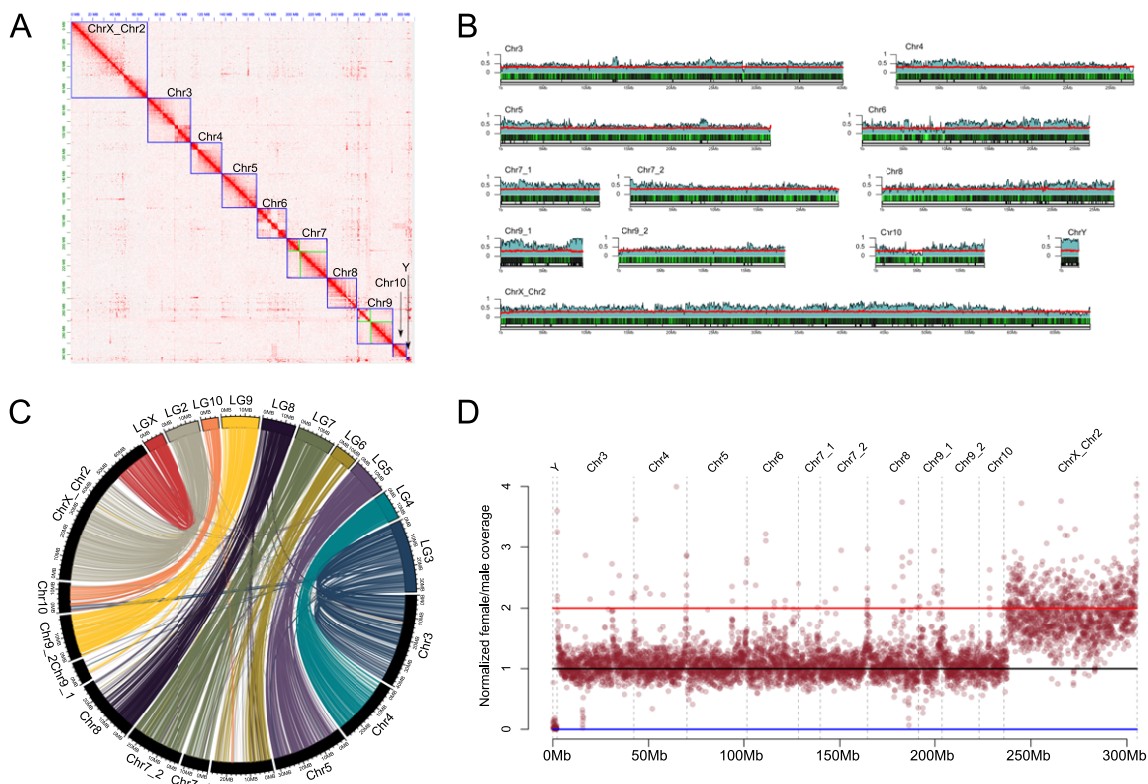

**Fig 2.** *Tribolium confusum* **genome assembly.** A) Hi-C heatmap showing long range contacts used to scaffold the genome assembly. Blue boxes denote putative chromosomes. Green boxes denote scaffolds placed within a chromosome that could not confidently be oriented (e.g., Chr7_1 and Chr7_2). B) Karyotype plot of the assembled chromosomes and shown from bottom to top, the length in Mb (grey), the scaffolding stitchpoints (vertical lines), gene density in 50 kb windows (black/green heatmap), the percentage of 50 kb window that was estimated to be repetitive (blue), and the percentage GC (red). C) Synteny comparison between *Tribolium confusum* and *Tribolium castaneum* (LG2-LGX) using 1:1 orthologs color coded by *T. castaneum* LG. Note the clear homology of *T. confusum* ChrX_Chr2 with LGX and LG2 of *T. castaneum*. D) Normalized female/male whole genome sequencing coverage over the *T. confusum* genome assembly. Each point represents a 50 kb window. Blue, black, and red horizontal lines indicate expected coverage over Y-linked, autosomal, and X-linked scaffolds, respectively. Note the uniform and female-biased coverage over the entire ChrX_Chr2 chromosome.

**Table 1. Draft genome assembly of *Tribolium confusum*.**

| Linkage group | Length (bp) | Contigs | Genes | Repetitive (%) |
|---|---|---|---|---|
| Chr2_ChrX | 69,141,766 | 98 | 2,839 | 51.2 |
| Chr3 | 40,139,663 | 29 | 1,950 | 45.7 |
| Chr4 | 27,691,043 | 19 | 1,488 | 44.0 |
| Chr5 | 31,655,641 | 29 | 1,737 | 46.3 |
| Chr6 | 26,721,382 | 61 | 1,399 | 51.6 |
| Chr7_1 | 11,552,149 | 19 | 442 | 56.2 |
| Chr7_2 | 24,417,818 | 11 | 1,451 | 43.1 |
| Chr8 | 27,266,789 | 74 | 1,581 | 47.2 |
| Chr9_1 | 9,615,367 | 97 | 240 | 67.0 |
| Chr9_2 | 19,562,745 | 16 | 1,283 | 40.2 |
| Chr10 | 12,905,899 | 16 | 746 | 41.0 |
| ChrY | 2,101,760 | 35 | 24 | 87.6 |
| Unplaced | 2,361,448 | 158 | 51 | 69.4 |

**Table 2. Repeats in the *Tribolium confusum* genome assembly.** Only those that represent > 1% of total genomic sequence shown.

| Repeat | Number of elements | Percentage of genome |
|---|---|---|
| Lines | 48,403 | 4.2% |
| LTR elements | 16,013 | 3.2% |
| DNA transposons | 88,450 | 10.3% |
| Unclassified | 468,963 | 30.3% |

assembly recovers the majority of chromosomes as a single scaffold (Fig 2A and 2B), and chromosome synteny is highly conserved between *T. confusum* and *T. castaneum* (Fig 2C). Note that we also assembled over 2 Mb of the repeat-rich Y chromosome and annotated 24 putative Y-linked genes (Table 1; see S1 Note for some analysis on these genes).

## Neo-sex chromosomes in beetles

*T. confusum* contains a neo-sex chromosome, that is, an autosome fused to the ancestral sex chromosome [5] and the neo-X chromosome of *T. confusum* is homologous to linkage group 2 of *T. castaneum* (Fig 2C). Neo-X and neo-Y chromosomes are initially homologous and diverge over time; the neo-Y chromosome degenerates and eventually loses nearly all homology with the neo-X [14]. Male and female genomic coverage analysis can be used to identify sex chromosomes and assess the level of homology between diverging sex chromosomes [9]. We find that genomic coverage, both for Illumina and Nanopore data, is about half for both the neo-X and the ancestral portion of the X chromosome in males compared to autosomes (Figs 2D and S1), indicating that the neo-sex chromosomes are highly differentiated in *T. confusum*. We see evidence of gene shuffling within the neo-X and ancestral X portions of the *T. confusum* X chromosome but not between neo-X and ancestral X regions (S2 Fig). This suggests that a distinct boundary exists between the ancestral and neo-X (such as a centromere) that prevents shuffling of genes among those segments of the X chromosome (S2 Fig).

To further explore the evolution of neo-sex chromosomes in beetles, we investigated the mountain pine beetle *D. ponderosae* [18], where females contain twelve pairs of chromosomes (11 autosomes, and a large neo-X chromosome) [4]. Importantly, the ancestral part of the neo-X/X chromosome (i.e., the part homologous to the X in *T. castaneum*) shows half the coverage in males compared to females (Fig 3B); the neo-X (that is, the part homologous to LG2 and LG4 of *T. castaneum*), however, only shows a modest coverage reduction in males and is heterogenous across its length (Fig 3B). Thus, there is still considerable homology between the neo-X and neo-Y of *D. ponderosae*, indicating that the neo-sex chromosomes of the mountain pine beetle are likely considerably younger than that of the confused flour beetle.

## Identification of Coleopteran "Stevens" elements

*D. ponderosae* (Dpon) and *T. castaneum* (Tcas) split over 200My ago [10,11], yet show high levels of conservation of linkage groups (Fig 3A). To investigate the level of conservation of chromosome homologies across beetles, we utilized published chromosome-level assemblies from four additional beetle species: *Pogonus chalceus* (Pcha), *Photinus pyralis* (Ppyr), *Propylea japonica* (Pjap) and *Chyrsomela aeneicollis* (Caen). These species span across the major phylogenetic branches of the beetle phylogeny and allow us to assess broad levels of chromosome homology across beetles (S2 Table). When identifying homologs of *T. castaneum* genes in the assemblies of these other species, we find that the majority of genes from each linkage group map to only one or a few linkage groups in all other species (Figs 4A and S3). In particular, we

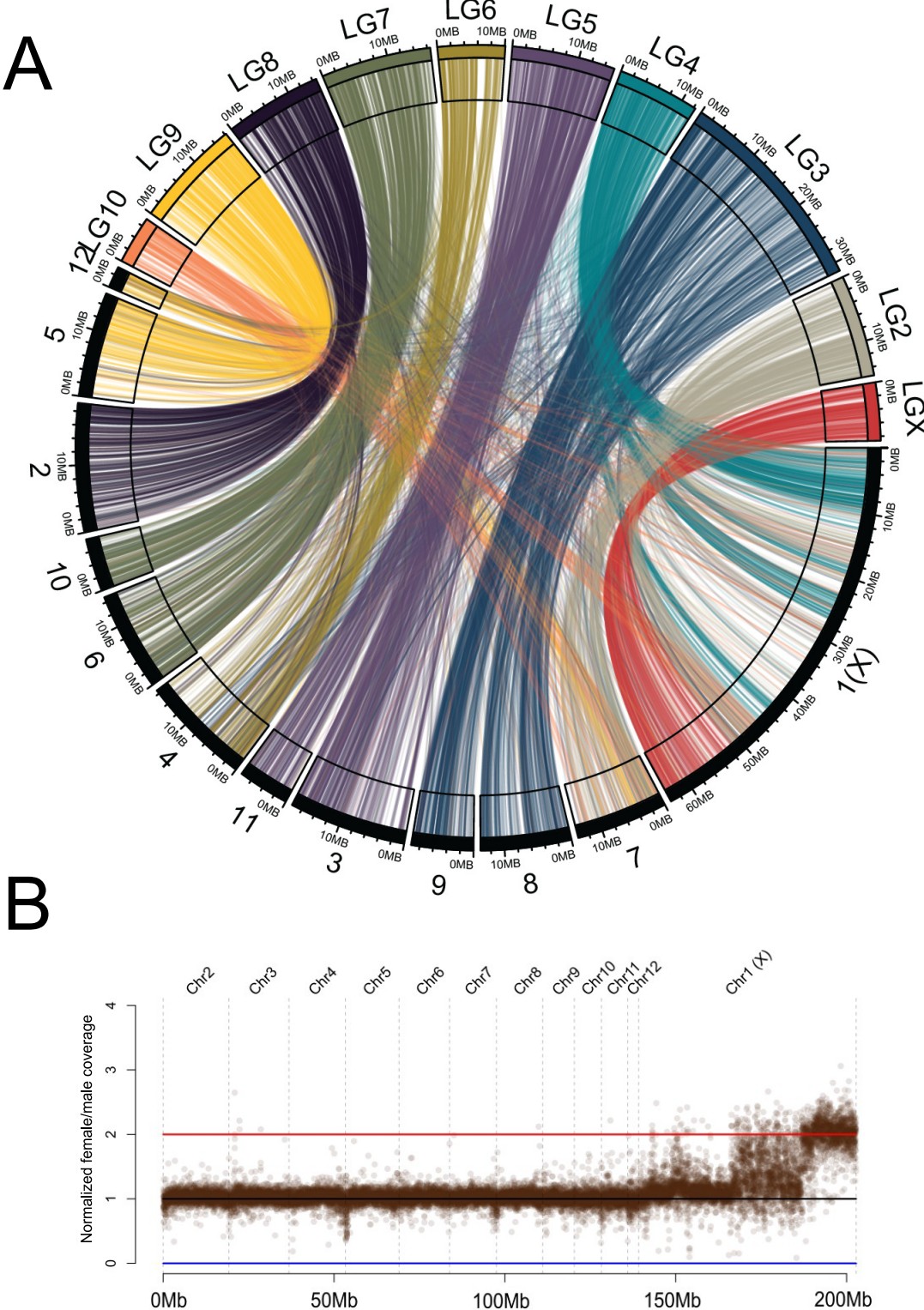

**Fig 3. Neo-sex chromosomes of the mountain pine beetle.** A) Synteny comparison between *Dendroctonus ponderosae* [1–12] and *Tribolium castaneum* (LG2-LGX) using 6,000 1:1 orthologs. The neo-X chromosome in *D. ponderosae* is comprised of regions homologous to *T. castaneum* LG4 and LG2 chromosomes. B) Normalized male and female whole genome sequencing coverage over the *D. ponderosae* genome assembly. Each point represents a 10 kb window.

find that the majority of genes on LG2-LG9 and the X chromosome of *T. castaneum* map to a single chromosome in at least one other species, suggesting that these might be ancestral linkage groups (Figs 4A, S3 and S4). Genes from LG10 of *T. castaneum* map to several different chromosomes in other beetles. Thus, we propose that beetles have similar ancestral linkage groups as have been identified for Diptera [termed Muller elements in Drosophila; 19], and more recently in nematodes [termed Nigon elements; 20]. We tentatively suggest that beetles have nine ancestral "Stevens elements" [Fig 4B; named after Nettie Stevens, whose pioneering work using the beetle *Tenebrio molitor* helped identify sex chromosomes; 21,22]. All other karyotypes of the beetles studied here can be explained by simple fusions and fissions of these ancestral Stevens elements (Table 3). Note, however, that other ancestral karyotype configurations are also possible, and detailed future studies of chromosome-level homologies of additional species spanning the beetle phylogeny should help to fine-tune the evolutionary reconstruction of ancestral karyotypes.

## Inference of sex chromosomes using male and female genomic coverage patterns

Our analysis of Coleoptera chromosome homology suggests that the ancestral X chromosome is conserved across beetles. To explore this in additional species spanning the Coleoptera phylogeny, we used male and female coverage data to confirm X-linkage of genes with chromosome level assemblies (see above) and also infer sex chromosomes in additional species with more fragmented genomes. We used published genome assemblies from four additional beetle

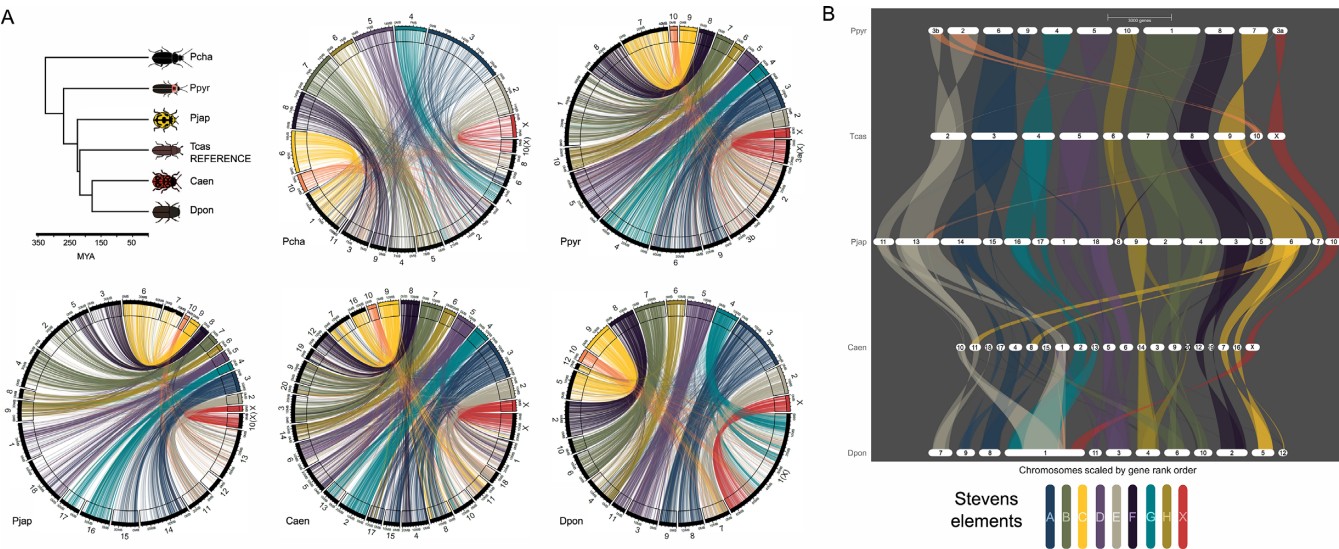

**Fig 4. Chromosome conservation in beetles.** A) Comparison of chromosome-scale assemblies of five Coleoptera spanning nearly 350 million years of evolution. All comparisons are between a focal species (Pcha = *Pogonus chalceus*, Ppyr = *Photinus pyralis*, Pjap = *Propylea japonica*, Caen = *Chrysomela aeneicollis*, Dpon = *Dendroctonus ponderosae*) and *Tribolium castaneum* (Tcas). Lines connect 1:1 orthologs and are color coded based on chromosomal location in *T. castaneum*. B) Synteny across five species from the suborder Polyphaga estimated using GENESPACE [60] and color coded by their chromosomal placement in *T. castaneum*. Stevens elements correspond to conserved linkage groups of the genome in beetles.

**Table 3. Proposed Stevens elements in beetles.** Shown is the number of linkage groups that make up an ancestral element with the assembly-specific linkage group numbering scheme in parentheses.

| Tcas LG | Stevens element | Pcha | Ppyr | Pjap | Caen | Dpon |
|---|---|---|---|---|---|---|
| LG3 | A | 1(6) | 2(6,9) | 2(14,15) | 4(4,8,15,17) | 2(8,9) |
| LG7 | B | 2(4,5) | 1(1) | 2(2,4) | 3(3,9,20) | 2(6,10) |
| LG9 | C | 1(1) | 1(7) | 2(6,7) | 2(7,16) | 2(5,12) |
| LG5 | D | 1(2) | 1(5) | 2(1,18) | 2(5,6) | 2(11,3) |
| LG2 | E | 1(8) | 2(2,3b) | 2(11,13) | 4(1,10,11,18) | 2(7,1) |
| LG8 | F | 1(9) | 1(8) | 2(3,5) | 2(12,19) | 1(2) |
| LG4 | G | 1(7) | 1(4) | 2(16,17) | 2(2,13) | 1(1) |
| LG6 | H | 1(5) | 1(10) | 2(8,9) | 1(14) | 1(4) |
| LG10 | - | 0 | 0 | 0 | 0 | 0 |
| LGX | X | 1(10) | 1(3a) | 1(10) | 1(X) | 1(1) |

species spanning the Coleoptera radiation (*Leptinotarsa decimlineata* (Ldec) [23]; *Anoplophora glabripennis* (Agla) [11]; *Agrilus planipennis* (Apla) and *Nicrophorus vespiloides* (Nves) [24]), and collected male and female genomic data, to identify which scaffolds are X-linked (Fig 5A and S2 Table). We identify 208, 32, 77 and 127 X-linked scaffolds in Ldec, Agla, Apla and Nves, respectively, based on female vs. male genomic coverage. Comparison of the location of annotated genes on these scaffolds with *T. castaneum* suggests that the vast majority of X-linked genes in these species correspond to genes found on LGX in *T. castaneum* (Fig 5B). Thus, the identity of the ancestral sex chromosomes appears widely conserved across beetles.

## Independent formation of neo-sex chromosomes in *Dendroctonus*

The western pine beetle, *D. brevicomis*, another member of the genus *Dendroctonus*, also has a neo-sex chromosome [25]. Comparative cytogenetic analysis suggests that neo-sex chromosomes may have evolved independently multiple times in *Dendroctonus*: *D. brevicomis* and its close relative *D. adjunctus* both have a large X and Y chromosome, indicative of a neo-sex chromosome and their sister species *D. approximatus* also has neo-sex chromosomes (Fig 6A). Their outgroups *D. mexicanus* and *D. frontalis*, on the other hand, have small X and tiny Y chromosomes (termed 'yp') [26]. The sister species to this clade, *D. ponderosae* and *D. jeffreyi*,

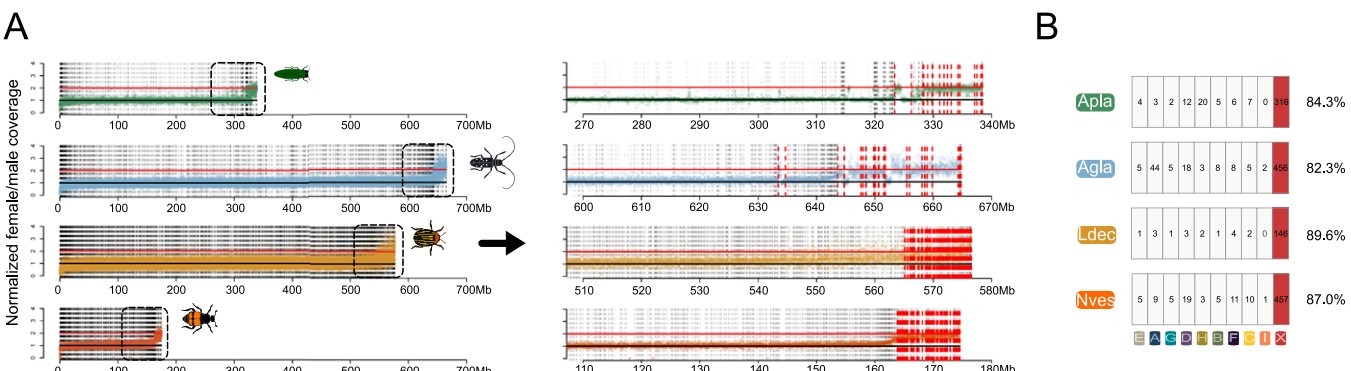

**Fig 5. Identifying X-linked scaffolds in fragmented beetle genome assemblies.** A) Normalized female/male whole genome sequencing coverage over contigs and scaffolds from four beetle genome assemblies. Dashed vertical lines separate contigs and scaffolds which were ordered left to right by increasing median female bias. Black and red horizontal lines denote expected autosomal or X-linked coverage, respectively. Right, shows the highlighted region (dashed boxed left) with putative X-linked scaffolds shown in red. B) Shown are the locations of putative X-linked 1:1 orthologs between the focal species and *Tribolium castaneum*.

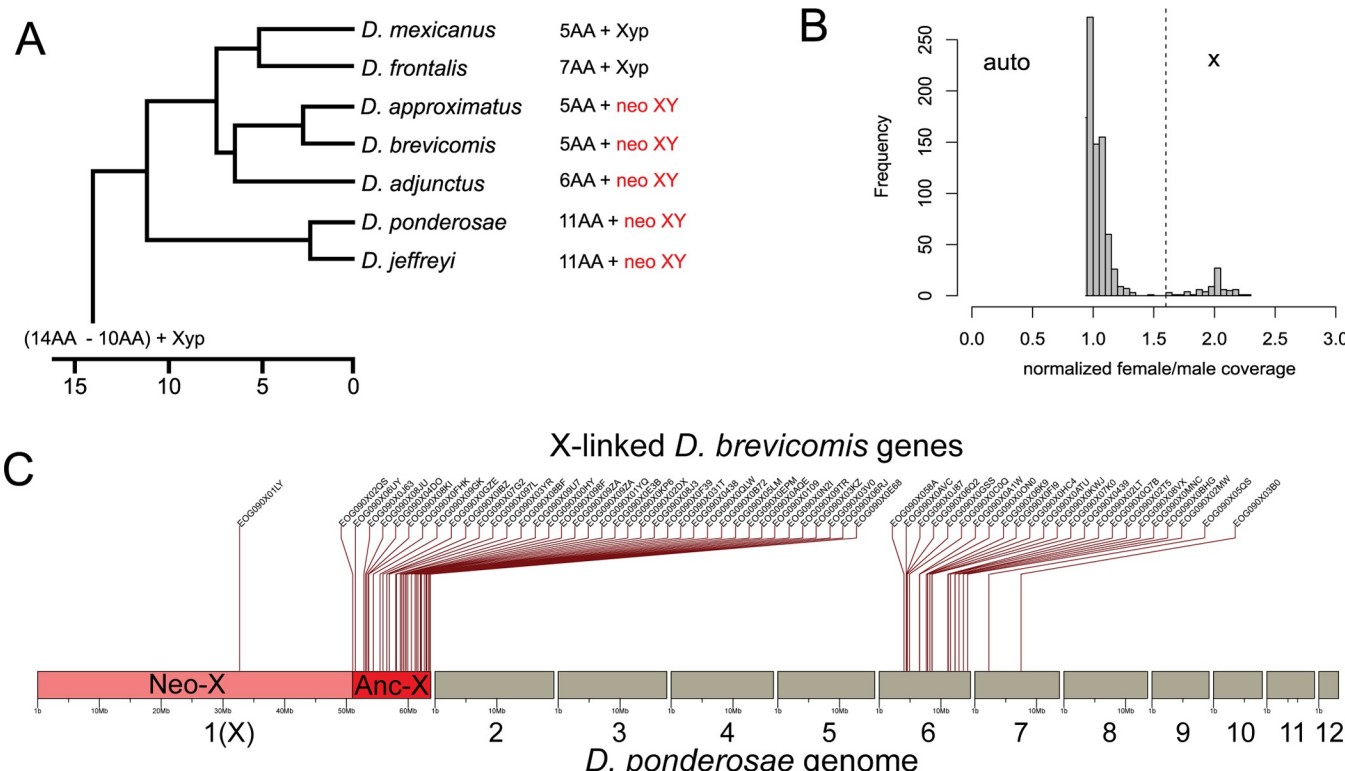

**Fig 6. Independent evolution of neo-sex chromosomes in *Dendroctonus* beetles.** A) Phylogenetic relationships of *Dendroctonus* beetles (adapted from [64]) with karyotypic formula shown to the right and neo-sex chromosomes highlighted in red. Karyotype squash images are from [25]. B) Normalized female/male whole genome sequencing data over scaffolds from the *D. brevicomis* genome assembly [27]. C) Shown are the genomic locations of *D. brevicomis* X-linked BUSCOs in the *D. ponderosae* genome assembly.

both harbor neo-sex chromosomes [4]. The phylogenetic distribution of neo-sex chromosomes thus suggests one of two scenarios: 1) a single chromosomal fusion event occurred in the ancestor to all these species and the neo-Y degenerated in *D. mexicanus* and *D. frontalis* to resembles a yp while in close relatives it has remained large or 2) neo-sex chromosomes have evolved at least twice in this species group through independent fusions in these clades (Fig 6A). To identify which Stevens elements (or analogous *D. ponderosae* chromosomes) are sex-linked in *D. brevicomis*, we again collected male and female *D. brevicomis* genome data and used a published assembly for *D. brevicomis* [27] to identify sex-linked scaffolds based on F/M coverage data (Fig 6B). Single copy orthologs (BUSCOs) from X-linked scaffolds in *D. brevicomis* map to either element X (ancestral X) or element B, which corresponds to the autosomal scaffold 6 in *D. ponderosae* (Fig 6C). Thus, our molecular analysis confirms that the neo-sex chromosome fusions happened independently in *D. brevicomis* and *D. ponderosae*, and a different ancestral linkage group became a neo-sex chromosome in these species. In addition, genomic coverage also suggests that the neo-sex chromosome fusion is much older in the western pine beetle (Dbre). Unlike in mountain pine beetle (see Fig 3), we find that coverage of element B genes (Dpon chr 6) in *D. brevicomis* is substantially reduced in males relative to females (Fig 6B), suggesting that there is little homology left between the neo-X and neo-Y chromosomes. Thus, neo-sex chromosomes evolved independently in *Dendroctonus*, and different ancestral elements fused to the ancestral X chromosome at different time points.

## Dosage compensation in beetles

In several taxa with differentiated sex chromosomes, dosage compensation mechanisms have evolved. Chromosome-wide dosage compensation mechanisms occur in mammals and many insects but appear absent in birds or monotremes [28,29]. Dosage compensation can be achieved in two steps, by globally up-regulating the X in both sexes, followed by inactivation of one X in females (like in mammals), or in one step by up-regulation of the X specifically in males (such as in Drosophila). Dosage compensation has been assessed in Tcas, and yielded controversial results: An initial study reported dosage compensation in males (i.e. similar expression of X-linked and autosomal genes in males), but hyper-transcription of the X in females compared to males [17]. This was interpreted as beetles following the 2-step model of dosage compensation of mammals, with only the first step having evolved. Follow-up studies confirmed dosage-compensation of the X in males but found no evidence of over-expression of the X in females [10,16], but a lack of dosage compensation in testis [16].

We took advantage of the neo-sex chromosomes in *T. confusum* to study how quickly dosage compensation evolves in beetles, and whether ancestral expression levels of the neo-X are restored in both sexes. We collected gene expression data from somatic tissues (male and female heads), as well as for ovaries and testes for *T. confusum* and compared them to published expression data in *T. castaneum*. Comparing X and neo-X expression to expression of autosomes will show whether both chromosomes have evolved dosage compensation and comparing the expression of the neo-X to its ancestral expression in *T. castaneum* (where it is autosomal) will reveal if ancestral expression levels are maintained. Contrasting male/female expression in somatic tissue suggests that both the ancestral X and the neo-X have evolved full dosage compensation (Fig 7A left). In testis, expression levels are similarly reduced relative to ovaries, both for the X/neo-X as well as the autosomes (Fig 7A right). Inferring ancestral expression patterns for neo-X genes suggests that dosage compensation restored ancestral expression levels of the neo-X in both sexes (Fig 7B). Thus, our analysis suggests that dosage compensation can evolve in response to neo-Y degeneration and restore ancestral expression levels of a sex chromosome in both sexes.

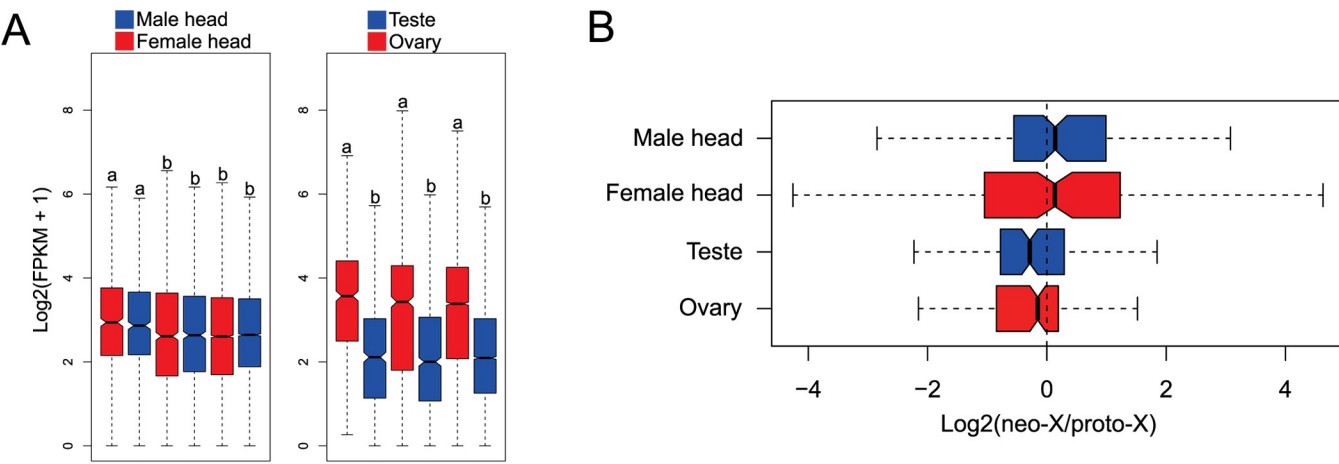

**Fig 7. Evolution of dosage compensation in *Tribolium confusum*.** A) Boxplots of male and female expression of ancestral X (677), neo-X (1,063), and autosomal (9,161) genes in the soma (heads). Only genes with FPKM ≥ 1 in either sex are shown. Expression of ancestral X (628), neo-X (958) and autosomal (8,187) genes in the gonads (testis and ovary). Only genes with FPKM ≥ 1 in either tissue shown. Boxes with different letters are significantly different from on another (*p* < 0.05, Bonferroni corrected Wilcoxon rank sum test). C) Male and female expression for neo-X vs. proto-X (i.e. ancestral autosome of Tcas) in soma and gonads. Outliers are not displayed on the graph.

## Evolution of sex-biased genes on sex chromosomes

In many taxa, genes with sex-biased expression show a non-random distribution on sex chromosomes. In *T. castaneum*, for example, genes with ovary-biased expression are enriched on the X chromosome while genes with testis-biased expression are depleted [16]. However, it is unclear how quickly ovary and testis-biased expression patterns evolve, and by what mechanism. That is, are ovary-biased genes more conserved across species, do existing X genes evolve ovary-biased expression, or do genes with ovary-biased expression preferentially move to the X? Conversely, testis-biased genes may be less conserved across species, genes located on the X chromosome may lose expression in testis or testis-biased genes may move off the X chromosome. And over how many million years do expression patterns change? To answer these questions, we first looked at conservation of X-linked genes across the five species with the most complete chromosome assemblies. We found 133 conserved 1:1 X-linked orthologs between Tcas, Ppyr, Caen, Pjap and Dpon, and their relative location along the chromosome has changed dramatically over time (Fig 8A). An evaluation of tissue-specific expression revealed that these genes show significant ovary-biased expression in multiple species and a deficiency of testis-biased expression (Fig 8B).

We also analyzed expression data for male and female somatic and gonad tissue in *T. confusum* and *D. ponderosae*. These two species have neo-sex chromosomes of different ages which enables us to examine the evolutionary dynamics of X-linked ovary and testis genes at different time scales. We see no striking differences in expression patterns in male and female heads across the ancestral and neo-X chromosome (Fig 8C and 8D), in both species. This suggests that the X is expressed similarly in the two sexes in somatic tissues, that is, the ancestral X and

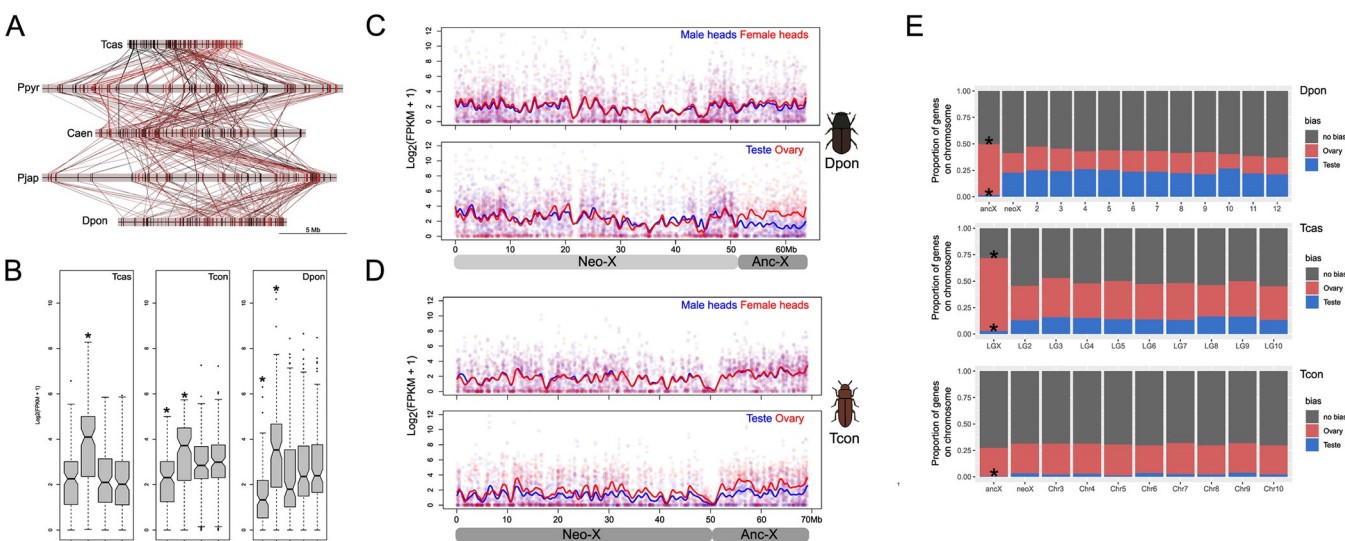

**Fig 8. Gene expression evolution on the X chromosome.** A) Chromosomal locations of 133 conserved X-linked orthologs in five chromosome-level assemblies. Each line represents a different protein coding gene. B) Boxplots showing expression of the 133 conserved X-linked genes across different tissues in Tcas, Tcon, and Dpon (* denotes tissues that significantly differed from all other tissues; $p < 0.05$, Bonferroni corrected Wilcoxon rank sum test). T = testis, O = ovary, gtM/F = gonadectomized male/female, hM/hF = head male/female. C) Expression of all 3,202 genes located on the X chromosome of Dpon (2,590 genes on the neo-X and 612 on the anc-X). The breakpoint between anc-X and neo-X (located at 52 Mb) was determined using a combination of sequencing coverage and chromosome synteny (Fig 3). A LOESS smoother was applied to show general trends in gene expression across the chromosome. D) Expression of all 2,839 genes located on the X chromosome of Tcon (1,932 genes on the neo-X and 907 on the anc-X). The breakpoint between anc-X and neo-X (located at 50 Mb) was determined using a combination of Hi-C and chromosome synteny (Fig 2). A LOESS smoother was applied to show general trends in gene expression across the chromosome. E) The proportion of gonadally sex-biased genes on the X chromosome and autosomes for Dpon, Tcas, and Tcon (* denotes significant enrichment or depletion based on hypergeometric tests ($p < 0.05$)). The neo-X and anc-X portion of the X chromosome were analyzed separately.

the older neo-X of *T. confusum* are fully dosage compensated, and both the neo-X and neo-Y alleles are still transcribed on the younger *D. ponderosae* neo-sex chromosome.

In both species, we find that the ancestral X is expressed at higher levels in ovaries, compared to testis. Thus, this is consistent with data from *T. castaneum*, showing enrichment of ovary-biased genes and depletion of testis-biased genes on the old, ancestral X. The neo-X, in contrast, shows different patterns depending on its age. The older neo-X of *T. confusum* overall shows higher expression in ovaries relative to testis, while no such difference is found on the younger neo-X of *D. ponderosae* (Fig 8C and 8D). Thus, this suggests that enough time has passed for sex-biased expression patterns to evolve towards the stereotypical patterns seen for X chromosomes on the older neo-X of *T. confusum*, but not yet for *D. ponderosae*.

We find that the ancestral X chromosome in *T. castaneum* is enriched for ovary-biased genes and depleted of testis-biased genes (Fig 8E), consistent with previous analyses [16]. We find that the ancestral X in *D. ponderosae* is also enriched for ovary-biased genes and depleted of testis-biased genes (Fig 8E). In *T. confusum*, the ancestral X is depleted of testis-biased genes but does not show a significant enrichment of ovary biased genes (Fig 8E). Thus, higher expression of the X in *T. confusum* ovary versus testis (Fig 8D) appears to be largely driven by a deficiency of testis-biased genes. Interestingly, the neo-X in both *T. confusum* and *D. ponderosae* shows no enrichment at all suggesting that neither neo-X chromosome has taken on some of the properties of 'old' X chromosomes.

## Conclusions

Our broad phylogenetic sampling of beetles suggests that they will be a promising model to study patterns of karyotype evolution, and evolutionary forces operating on sex chromosomes. We show that ancestral linkage groups are maintained across beetles which allows the study of karyotype evolution at the molecular level. We find that the ancestral X is conserved in Coleoptera, but identify multiple independent formations of neo-sex chromosomes, even among closely related beetle species. An abundance of beetles and further development of genomic and functional tools will make Coleoptera a valuable model system to study the molecular mechanisms and evolutionary forces driving the differentiation of sex chromosomes across taxa.

## Methods and materials

### Genome sequencing and assembly

The *Tribolium confusum* strain we sequenced was obtained from Heath Blackmon (Texas A&M University). We generated data for both sexes using high molecular weight DNA that was extracted from ~ 50 sexed pupae using a Qiagen Blood & Cell Culture DNA Midi Kit. We size selected DNA for fragments >15 kb using BluePippin (Sage Science). For size selection, we used 6 μg of total DNA (100 ng/μl) run in two wells. The elute was bead purified, resulting in a total of 2.7 μg total DNA in a 50 μl solution. We generated long-reads using Nanopore and the SQK-LSK109 sequencing kit on one 9.4.1RevD flow cell and with the minKNOW software version 3.1.13. FAST5 output files were basecalled using Guppy version 3.0.3 (Oxford Nanopore Technologies) with default quality score filtering.

We used a two-step process to assemble the genome. First, for each sex, we used Canu version 1.8 [30] to error-correct Nanopore sequencing reads using the following parameters (correctedErrorRate = 0.065 corMinCoverage = 8 batOptions = "-dg 3 -db 3 -dr 1 -ca 500 -cp 50" trimReadsCoverage = 4 trimReadsOverlap = 500 genomeSize = 200m). We then assembled all corrected reads from both sexes using WTDBG2 [31] with default settings. We then removed non-target sequence from the assembly by running BLAST on all contigs < 1 Mb (n = 158

contigs, median = 10 kb) against the nt database and returned the top two hits. Contigs with either no hit, or a hit to *T. castaneum*, were retained. After removing non-target sequence, we polished the assembly using our Nanopore reads and three rounds of Racon [32] and using Illumina data and three rounds of Pilon [33]. This method of combining multiple rounds of Racon and Pilon has been shown to increase genome assembly quality in Drosophila [34]. To polish with Racon we mapped our raw male and female nanopore reads each round using minimap2 and specified -x ava-ont [35]. For genome polishing with Pilon we used Illumina 100bp PE reads derived from a ChIP-seq experiment. During assembly and polishing we assessed genome completeness using BUSCO v3 [36] and the odb9 eukaryota database.

## Hi-C scaffolding

To scaffold the assembly, we used chromatin conformation capture to generate Hi-C data [37]. Hi-C libraries were generated as outlined in Bracewell et al. [38] using a DNase digestion method [39]. The resulting DNA library was prepped using an Illumina TruSeq Nano library prep kit and was sequenced on a HiSeq 4000 with 100bp PE reads. We used the Juicer pipeline [40] to map Hi-C reads and generate contact maps based on 3D interactions. We then used the 3D-DNA pipeline [41] and default settings with no iterative rounds for misjoin correction (-r 0) and in haploid mode to scaffold the contigs. 3D-DNA output files were visualized and checked for accuracy using Juicebox [42] with verification and modifications to scaffolding done using built-in tools based on recommendations (https://aidenlab.org/assembly/manual_180322.pdf). The final assembly was scaffolded together with 300 Ns between each contig.

## Repetitive element identification and genome masking

We used two methods, EDTA [43] and RepeatModeler2 version 2.0.5 [44], to identify repetitive sequence in the genome assembly. With these repeat libraries we then masked the genome with RepeatMasker version 4.1.5 and rmblastn version 2.13.0 using the -no_is and -nolow flags. We used the default settings and output table to identify repeat enrichment. To characterize the proportion of sequence and genomic distribution of repetitive elements we used bedtools nuc and coverage [45].

## Genome annotation and characterization of assembly

To annotate the *T. confusum* genome assembly we used the masked genome (above) and the MAKER annotation pipeline [46] to identify gene models. The *ab initio* gene predictors SNAP [47] and Augustus [48] were used to guide the annotation along with the protein set from *T. castaneum*. To help in annotation, we used Hisat2 [49] and the -dta flag to align RNA-seq data (described below) to the draft genome and StringTie2 [50] to build a *Tribolium confusum* transcriptome. Transcripts were created from the transcriptome file using gffread. For gene prediction with SNAP we used the *D. melanogaster* hmm and for Augustus we used the *T. castaneum* beetle model. We set the MAKER parameters est2genome and protein2genome to 1 to allow the creation of gene models from the *T. confusum* transcriptome alignment and *T. castaneum* protein alignment. KaryoploteR [51] was used to generate to plots characterizing general features of the genome assembly.

## Neo-sex chromosomes in beetles

We used differences in read mapping between males and females to explore the level of differentiation between the neo-X and neo-Y in *T. confusum* and *D. ponderosae*. For *T. confusum*, we used our draft genome assembly (above) and mapped our nanopore reads using minimap2

[35] and mapped previously published Illumina reads (SRA: SRX13185890 and SRX13185891) using BWA MEM [52]. For *D. ponderosae*, we used the published chromosome-level assembly [18] and previously published Illumina data for a male and female (SRA: SRX3381432 and SRX3381430).

## Coleoptera chromosome synteny comparison

Our analysis focused on six species with chromosome level assemblies that span the large Coleoptera phylogeny. Those species include: *Pogonus chalceus* [53], *Propylea japonica* [54], *Chrysomela aeneicollis* [55], *Photinus pyralis* [56], *Tribolium castanem*, *Tribolium confusum* (present study), and *Dendroctonus ponderosae* [18]. Our initial analysis of *P. japonica* revealed that the assembly deposited in GENBANK was not consistent with the publication [54] and so we re-scaffolded with the published Hi-C data and created our own more conservative linkage groups (S5 Fig). Briefly, we took the published assembly and split all scaffolds and then used methods for mapping Hi-C data and scaffolding an assembly described above. To compare genome assemblies with *T. castaneum* (version 5.2), we used protein sets from each species in our analysis and orthoDB [57] and default settings to identify 1:1 orthologs. We used the longest predicted protein sequence from all protein coding genes. We used R and *circlize* [58] and RIdeogram [59] to produce synteny plots. To identify putative Stevens elements in beetles, we used GENESPACE [60] and the protein sets from above. An initial run using default settings identified chromosome synteny, but low levels of collinearity hampered comparisons between some divergent genomes. We therefore adjusted parameters to blkSize = 3, blkRadius = 250, and synBuff = 200. Associated GENESPACE dotplots showing all pairwise comparisons are available in Figs S6–S15.

## Identifying X-linked scaffolds in *Dendroctonus brevicomis*

We collected *D. brevicomis* from near Dutch Flat, California, extracted DNA using a Qiagen DNeasy kit, and whole genome Illumina sequenced a male and a female (as above). We used the *D. brevicomis* genome assembly, which is comprised of 35,469 scaffolds with an N50 scaffold length of 5 kb (maximum = 540,064bp). We mapped male and female *D. brevicomis* Illumina data to the reference genome using BWA MEM and default settings and estimated the median coverage over all contigs in the assembly. The male was sequence to ~33x and the female was sequence to ~23x coverage. We investigated only complete or fragmented BUSCOS (937 total). Of those, a total of 74 were found on scaffolds with $\geq$ 1.6x normalized female/male coverage (calculated by median coverage per scaffold). These 57 putative X-linked scaffolds have a combined length of 5,659,142 bp.

## Gene expression and gene enrichment analyses

To characterized genomic patterns of gene expression, we first generated RNA-seq data for *Tribolium confusum*. We used the same line that was used for the genome assembly, and we isolated total RNA from male and female heads, and ovaries and testes of adults. Tissues were dissected in PBS and quickly transferred to TRIZOL for preservation and extraction. We used the Illumina TruSeq Stranded RNA kit to prepare libraries that were subsequently sequenced on an Illumina HiSeq 4000 with 100 bp PE reads. For analysis of gene expression in *T. castaneum* and *D. ponderosae* we re-analyzed RNA-seq data generated from sex-specific tissues [16,61]. In all analyses we used Hisat2 [49] to align the RNA-seq data and StringTie2 [50] with default settings to estimate gene expression levels (FPKM). We estimated median expression of the two replicates per tissue type (ovary, testis, gonadectomized male, gonadectomized female) in *T. castaneum*. In *D. ponderosae*, where the replicates were often restricted to certain

tissues (e.g., Female head replicate 1 and 2), we confirmed results were consistent regardless of replicate. To test for enrichment or depletion of gonad-biased genes, we identified gonad-biased genes as those with gene expression values (FPKM) > 2 fold higher than all other tissues (e.g.,[23]). To test for enrichment or depletion of testis or ovary biased genes on a particular chromosome, we used hypergeometric tests in R with the *phyper* package and accounted for multiple tests using a Bonferroni correction.

## Supporting information

**S1 Table. BUSCO results from each round of assembly and polishing of the draft genome for *Tribolium confusum*.**
(PDF)

**S2 Table. Description of the eleven genomes used in comparative analyses.**
(PDF)

**S1 Fig.** Mean female (top) and male (bottom) Nanopore sequencing coverage (log2) plotted along the *Tribolium confusum* genome assembly in 50 kb windows.
(PDF)

**S2 Fig. 1,621 1:1 orthologs of *Tribolium castaneum* LGX and LG2 and their relative locations on the fused *Tribolium confusum* ChrX_Chr2 chromosome.**
(PDF)

**S3 Fig. Heatmaps showing locations of 1:1 orthologs in genome comparisons with other beetle species with chromosome-level assemblies.** Putative Stevens elements are shown on the horizontal (A-H, X) while species-specific naming schemes for linkage groups from each draft genome assembly are shown on the vertical. Note that LG10 in *Tribolium castaneum* (salmon) is shown but was not found to be conserved over time and is not considered a Stevens element.
(PDF)

**S4 Fig. Chromosome conservation in beetles.** Synteny across five species from the suborder Polyphaga using 2,244 1:1 orthologs color coded by their chromosomal placement in *Tribolium castaneum*. Stevens elements correspond to conserved linkage groups of the genome in beetles.
(PDF)

**S5 Fig.** A) Hi-C contact map showing the published scaffolded *Propylaea japonica* genome assembly (Linkage groups 0–9 from left to right). We found widespread within-linkage group scaffolding that was not supported by the Hi-C data. B) Our re-scaffolded assembly with linkage groups shown on the vertical (right). Our naming scheme follows the original LG name from the published assembly followed by additional notation when we were unable to confidently orient into one linkage group. For example, LG1_1 and LG1_2 represent two scaffolds that originate from the published LG1. LG1_1 and LG1_2 likely represent two arms of a single chromosome.
(PDF)

**S6 Fig. GENESPACE dotplot between *Chyrsomela aeneicollis* (Caen) and *Dendroctonus ponderosae* (Dpon).**
(PDF)

**S7 Fig. GENESPACE dotplot between *Tribolium castaneum* (Tcas) and *Dendroctonus ponderosae* (Dpon).**
(PDF)

**S8 Fig. GENESPACE dotplot between *Chyrsomela aeneicollis* (Caen) and *Propylea japonica* (Pjap).**
(PDF)

**S9 Fig. GENESPACE dotplot between *Dendroctonus ponderosae* (Dpon) and *Propylea japonica* (Pjap).**
(PDF)

**S10 Fig. GENESPACE dotplot between *Photinus pyralis* (Ppyr) and *Propylea japonica* (Pjap).**
(PDF)

**S11 Fig. GENESPACE dotplot between *Tribolium castaneum* (Tcas) and *Propylea japonica* (Pjap).**
(PDF)

**S12 Fig. GENESPACE dotplot between *Chyrsomela aeneicollis* (Caen) and *Photinus pyralis* (Ppyr).**
(PDF)

**S13 Fig. GENESPACE dotplot between *Dendroctonus ponderosae* (Dpon) and *Photinus pyralis* (Ppyr).**
(PDF)

**S14 Fig. GENESPACE dotplot between *Tribolium castaneum* (Tcas) and *Photinus pyralis* (Ppyr).**
(PDF)

**S15 Fig. GENESPACE dotplot between *Chyrsomela aeneicollis* (Caen) and *Tribolium castaneum* (Tcas).**
(PDF)

**S1 Note. Y-linked gene analysis.**
(PDF)

## Acknowledgments

We would like to thank Heath Blackmon for sending us samples.

## Author Contributions

**Conceptualization:** Ryan Bracewell, Doris Bachtrog.

**Data curation:** Ryan Bracewell, Doris Bachtrog.

**Formal analysis:** Ryan Bracewell, Anita Tran.

**Funding acquisition:** Doris Bachtrog.

**Investigation:** Ryan Bracewell, Anita Tran, Kamalakar Chatla.

**Methodology:** Ryan Bracewell, Doris Bachtrog.

**Project administration:** Doris Bachtrog.

**Resources:** Anita Tran, Kamalakar Chatla, Doris Bachtrog.

**Supervision:** Doris Bachtrog.

**Visualization:** Ryan Bracewell.

**Writing – original draft:** Ryan Bracewell, Doris Bachtrog.

**Writing – review & editing:** Ryan Bracewell, Doris Bachtrog.

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
