## [Decision Letter · Decision Letter 0]

20 Mar 2024

Dear Dr Bachtrog,

Thank you very much for submitting your Research Article entitled 'Sex and neo-sex chromosome evolution in beetles' to PLOS Genetics.

The manuscript was fully evaluated at the editorial level and by independent peer reviewers. The reviewers appreciated the attention to an important topic but identified some concerns that we ask you address in a revised manuscript.

The reviewers agreed that beetle sex chromosomes have been understudied and this is an important contribution to the field. 

The manuscript is written clearly and there were no concerns with your overall interpretation of the results. The reviewers had some recommendations to improve the overall clarity of the manuscript. In particular, there were some areas of the methods that were not adequately described. In addition, two reviewers were concerned with the accessibility of your datasets. It appears the resequencing, genome assembly, and RNA-seq data are available under BioProject PRJNA715483. Please make sure all datasets are publicly available here. 

The reviewers also suggested some additional analyses with the genes on the Y chromosome. The total number of genes on the Y chromosome were only mentioned in Table 1. It would helpful to explore these genes further. In other species, genes on the Y chromosome can be enriched for male specific functions and can have interesting patterns of selection. In T. confusum, are the 24 remaining genes enriched for any particular functions? Are these ancestral genes?  In addition, are any of these genes under any form of selection? 

We therefore ask you to modify the manuscript according to the review recommendations. Your revisions should address the specific points made by each reviewer.

Yours sincerely,

Michael White

Guest Editor

PLOS Genetics

Justin Fay

Section Editor

PLOS Genetics

Reviewer's Responses to Questions

**Comments to the Authors:**

Reviewer #1: This is a nice study examining the evolutionary history of sex chromosomes in beetles. I really have only minor queries/items of clarity, and one suggestion for complementary data analysis.

1) There doesn't seem to be any mention of the structure and gene composition of the assembled Y, other than its presence in Table 1. This is surprising. What genes are remaining on the Y, are there spatial patterns of loss, and are there interesting patterns of gene retention/loss by gene function?

2) I would recommend the authors try GeneSpace as an alternative/complement to identify and visualize syntenic orthologues across these genomes, I think it can help provide some clearer patterns of visualization across species. (https://github.com/jtlovell/GENESPACE). I find the decisions around the "Stevens element' designation to be semi-arbitrary from the synteny plots shown, but perhaps it would be more convincing with Genespace plots, and/or some additional justification of how these were decided on in the methods. But great tribute to Nettie Stevens with the naming!

Reviewer #2: This is the first study to comprehensively reconstruct sex chromosomes, their characteristics, and evolution in detail in a large number of beetle species. Beetles are a very specious group with potentially numerous independent origins of new sex chromosomes and so represent a very valuable model to study the origin and evolution of sex chromosomes and sex determination. However, as yet they have received limited attention in this regard. This elegant study generates genomic and expression data for several species and combines this will publicly available data across eleven species. They identify significant patterns of sex chromosome evolution and open up new avenues for future research. The paper is well written and easy to follow for a broad audience and the figures are well presented. I have only minor comments.

Methods are lacking detail in places. I would expect to see a Supplementary Methods section for these types of analyses.

L268 Can the authors include information on the thresholds used to identify non-target sequences

L282 - 285 Further details on the thresholds used in the 3d-DNA pipeline are necessary

L296 How were SNAP and Augustus run

L319 Where did the D. brevicomis genome come from? Is it publicly available or assembled as a part of this paper?

Given the significant enrichment in repetitive elements, it would be helpful for readers to break this down by type of element in a new table.

The authors have made the genome assembly and raw reads available but not the RNA-seq data they generated for T. confusum or the resequencing data for D. brevicomis

L12 Missing comma after Tribolium confusum

L46 Unnecessary comma after high conserved

L58 Should be evolving

L60, L206 Numbered references Charlesworth & Charlesworth 2000 Bachtrog 2013 & WHITTLE et al. 2020

L63, 65 Need references to support these statements

L311 Typo should be and

L291 Typo should be characterize

L115 Should Figure 2C be referenced here?

Reviewer #3: The authors provide the most complete genomically informed examination of sex chromosome evolution in beetles to date. I believe this is an important contribution that will be of broad interest. I think there are a few things that could be added or adjusted to make the article even better prior to publication.

Line 50 This is more of a question for the authors. Each time I read places where you say, "chromosomal sex determination is conserved across beetles," I kept asking myself whether haplodiploidy is considered a form of chromosomal sex determination. I wonder if, in the introduction, it might be good to provide some basic information on the proportions of species with XY, XO, HD, and parthenogenetic systems. I think this would give a broader perspective and context for the study.

Line 57 It might be worth mentioning in the introduction that you are including species that have non-pairing sex chromosomes and pairing sex chromosomes and variation in this characteristic across beetles. The fact that you mention the yp (distant pairing parachute Y chromosome) notation for some sex chromosomes later (line 156) makes me feel more strongly that you should explain variation in pairing and whether you think this impacts the course of sex chromosome evolution in your Adephaga species vs the Polyphaga species that are dominated by distant pairing sex chromosomes.

Line 60 Citations are mixed in number format and author year

Line 94 The statement that the NeoY will lose all homology to the Neo X seems to be overly strong. Many Y chromosomes retain some subset of genes that are essential and have gametologs on the X. This seems to be too strong of a blanket statement.

Line 309 is this new assembly deposited in a repository?

**Have all data underlying the figures and results presented in the manuscript been provided?**

Reviewer #1: Yes

Reviewer #2: **No: **The authors have made the genome assembly and raw reads available but not the RNA-seq data they generated for T. confusum or the resequencing data for D. brevicomis

Reviewer #3: Yes

PLOS authors have the option to publish the peer review history of their article (what does this mean?). If published, this will include your full peer review and any attached files.

Reviewer #1: No

Reviewer #2: No

Reviewer #3: No

---

## [Editor Report · Decision Letter 1]

30 Oct 2024

Dear Dr Bachtrog,

We are pleased to inform you that your manuscript entitled "Sex and neo-sex chromosome evolution in beetles" has been editorially accepted for publication in PLOS Genetics. Congratulations!

Yours sincerely,

Michael A White

Academic Editor

PLOS Genetics

Justin Fay

Section Editor

PLOS Genetics

Aimée Dudley

Editor-in-Chief

PLOS Genetics

Anne Goriely

Editor-in-Chief

PLOS Genetics

Comments from the reviewers (if applicable):

**Data Deposition**

http://datadryad.org/submit?journalID=pgenetics&manu=PGENETICS-D-24-00112R1

**Press Queries**

---

## [Editor Report · Acceptance letter]

8 Nov 2024

PGENETICS-D-24-00112R1 

Sex and neo-sex chromosome evolution in beetles 

Dear Dr Bachtrog, 

We are pleased to inform you that your manuscript entitled "Sex and neo-sex chromosome evolution in beetles" has been formally accepted for publication in PLOS Genetics! Your manuscript is now with our production department and you will be notified of the publication date in due course.

With kind regards,

Dorothy Lannert

PLOS Genetics

On behalf of:
